# KRAB-ZFP Transcriptional Regulators Acting as Oncogenes and Tumor Suppressors: An Overview

**DOI:** 10.3390/ijms22042212

**Published:** 2021-02-23

**Authors:** Joanna Sobocińska, Sara Molenda, Marta Machnik, Urszula Oleksiewicz

**Affiliations:** 1Department of Cancer Immunology, Poznan University of Medical Sciences, Rokietnicka 8, 60-806 Poznan, Poland; a.s.sobocinska@gmail.com (J.S.); sara.molenda97@gmail.com (S.M.); 2Department of Cancer Diagnostics and Immunology, Greater Poland Cancer Centre, 15 Garbary Street, 61-866 Poznan, Poland

**Keywords:** KRAB-ZFPs, cancer, epigenetic repressor, ZNF471, ZBRK1, ZKSCAN3, ZNF300, ZFP57, ZNF224

## Abstract

Krüppel-associated box zinc finger proteins (KRAB-ZFPs) constitute the largest family of transcriptional factors exerting co-repressor functions in mammalian cells. In general, KRAB-ZFPs have a dual structure. They may bind to specific DNA sequences via zinc finger motifs and recruit a repressive complex through the KRAB domain. Such a complex mediates histone deacetylation, trimethylation of histone 3 at lysine 9 (H3K9me3), and subsequent heterochromatization. Nevertheless, apart from their repressive role, KRAB-ZFPs may also co-activate gene transcription, likely through interaction with other factors implicated in transcriptional control. KRAB-ZFPs play essential roles in various biological processes, including development, imprinting, retroelement silencing, and carcinogenesis. Cancer cells possess multiple genomic, epigenomic, and transcriptomic aberrations. A growing number of data indicates that the expression of many KRAB-ZFPs is altered in several tumor types, in which they may act as oncogenes or tumor suppressors. Hereby, we review the available literature describing the oncogenic and suppressive roles of various KRAB-ZFPs in cancer. We focused on their association with the clinicopathological features and treatment response, as well as their influence on the cancer cell phenotype. Moreover, we summarized the identified upstream and downstream molecular mechanisms that may govern the functioning of KRAB-ZFPs in a cancer setting.

## 1. Introduction to Epigenetic Mechanisms Regulating Gene Transcription in Cancer

Cancer development comprises a variety of genetic and epigenetic events. Hanahan and Weinberg proposed a solid foundation for understanding tumor biology based on the characteristic cancer hallmarks [1,2]. Tumor cells, among other features, may remain continuously responsive to proliferative signals, resist cell death, induce angiogenesis, activate invasion and metastasis, and evade immunological surveillance. Gradual acquisition of tumor-related traits by a single cell is interwoven with progressive genome and epigenome instability that further facilitates the selection of the fittest clone giving rise to cancer. Genetic causes of oncogenesis are well studied and are considered the main force of tumor transformation. They lead to the loss, amplification, changed sequence or structure of a number of genes, thus altering the amount or functioning of the proteins that may drive carcinogenic processes. Indeed, a large number of signaling pathways (e.g., p53, Wnt/β-catenin, MAPK, NF-κB pathways) may be modified in a malignant cell, consequently affecting all the proposed hallmarks of cancer [1,2].

Although the genetic events are well-defined factors responsible for cancer development, rapidly expanding knowledge about cancer epigenetics allows unraveling the role of driver aberrations within cancer epigenome. Different stimuli involving genetic, environmental, and metabolic events influence the balance of the chromatin state, which in turn may become more restrictive or permissive. Tight and repressive chromatin structure may cause the blockade of cellular transcription programs, whereas a more permissive state enhances epigenetic and transcriptional plasticity. Increased plasticity enables malignant or premalignant cells to acquire cancerous features in a stochastic manner, thus increasing heterogeneity within tumor mass [3].

Many epigenetic states and signals are largely distorted in cancer, and an altered DNA methylation profile is the best-studied example of this phenomenon [4]. In general, cancer cells may be characterized by global DNA hypomethylation and focal hypermethylation. Loss of DNA methylation is frequently noted within gene promoters, enhancers, non-coding sequences, and repeated elements [5,6], which triggers overexpression of several oncogenes [7,8,9], imprinted genes [10,11,12,13,14], cancer-testis genes [5,15], and transposable elements (TEs) [16,17,18,19,20]. In contrast, tumor suppressor genes (TSGs), which are involved in cell proliferation, apoptosis, DNA repair, or immune response, are frequently silenced by the hypermethylation within gene promoter [4,21].

Other epigenetic events, such as post-translational histone modification (PTMs), chromatin 3D organization, and nucleosome positioning, are also dysregulated in cancer cells. Global analyses revealed that chromatin-modifying enzymes are frequently altered in multiple tumors [22]. Interestingly, proteins establishing repressive histone marks tend to be upregulated, whereas those responsible for the deposition of activating marks have lower expression and recurrent mutations [22]. Altogether, variations within epigenetic machinery influence the expression of other genes, mutation rates, and genome stability [4]. A specific group of epigenetic repressors—Krüppel-associated box domain zinc finger proteins (KRAB-ZFPs) is commonly deregulated in cancer cells [23,24]. Different members of the family tend to be over- or underexpressed in distinct tumors. To date, numerous reports indicate that specific factors influence various aspects of carcinogenesis. This review will discuss known examples of KRAB-ZFPs acting as oncogenes or TSGs, summarize their influence on cancer biology, and present molecular mechanisms responsible for their function.

## 2. An Overview on KRAB-ZFP Superfamily Genes: Origin and Structure

KRAB-ZFPs constitute the most numerous family of transcriptional regulators and epigenetic repressors found in higher vertebrates [25,26]. The appearance of KRAB-ZFP genes is estimated to around 420 million years ago, and their roots lead to a common ancestor of coelacanths, lungfish, and tetrapods [26,27]. The multitude of KRAB-ZFP genes in mammals arose due to repeated cycles of tandem duplication and subsequent diversification of an ancestor gene [28]. The human genome comprises over 400 KRAB-ZFP protein-coding genes. Most of them localize within 25 major clusters, with the largest one located on chromosome 19 (HSA19) [29,30]. Repetitive duplication processes are responsible for the divergence among KRAB-ZFPs: they show distinctive tissue-specific expression patterns and differences within DNA-binding domains [28,31].

The KRAB domain consists of approximately 50–75 amino acids located at the N-terminus of the protein and may be divided into subdomains or boxes (Figure 1). The A-box is responsible for the repressive activity of the domain by recruiting KRAB-associated protein 1 (KAP1, also referred to as TRIM28 (tripartite motif protein 28)). In contrast, the B-box is believed to enhance A-box interaction efficiency [26,32]. A B-box within the KRAB domain is not obligatory, and A-box may be found alone. Some of the KRAB-ZFPs also contain additional domains at their N-terminus, such as “SRE-ZBP, CTfin51, AW-1 and Number 18 cDNA” (SCAN) or “domain of the unknown function” (DUF3669). The SCAN domain mediates homo- and heterodimerization between different SCAN-domain proteins, while the function of the DUF3669 domain remains poorly characterized (Figure 1) [26,32,33].

## 3. KRAB-ZFP/KAP1 Duo as a Potent Epigenetic Repressor Complex

The canonical function of KRAB-ZFP factors is related to the epigenetic repression of various chromatin regions, including gene promoters and 3’ ends, imprinted control regions, as well as repeat elements. KRAB-ZFPs exert their epigenetic silencing role by recruiting KAP1 protein [25,34,35]. The KRAB-ZFP/KAP1 complex activates chromatin remodeling by assembling a histone-modifying complex (Figure 2) containing several epigenetic factors [i.e., Nucleosome Remodeling Deacetylase (NuRD) —histone deacetylase-containing complex, SET Domain Bifurcated 1 (SETDB1)—H3K9me3-specific histone methyltransferase, and HP1—heterochromatin protein 1]. As such, KRAB-ZFP/KAP1 interaction is essential for histone deacetylation, H3K9 trimethylation, and heterochromatization within a genetic locus recognized by the zinc fingers of a given KRAB-ZFP [35,36,37].

KAP1 contains multiple domains that confer the interaction with other proteins or possess enzymatic activity. For example, the RBCC domain located within KAP1 N-terminus is responsible for the interaction with the KRAB domain, forming the repression complex as described above [38]. Moreover, the RING finger domain of KAP1 may be involved in tumorigenesis due to its ubiquitin E3-ligase activity, which results in the ubiquitination and proteasomal degradation of p53 and AMPK [39]. Of note, the multi-domain structures of KRAB-ZFPs and KAP1 render the duo highly flexible, as their role largely depends on their interactome within a particular cell. Indeed, KRAB-ZFPs were repetitively shown to exert co-repressors and co-activator functions, whereas KAP1 is known for its pleiotropic effects in various research models [25,39]. These observations highlight the complexity of the biological functions executed by KRAB-ZFP superfamily genes.

## 4. The Canonical Repressive Functions of KRAB-ZFPs in Normal Cells: Guarding Imprinted Regions and Transposable Elements

KRAB-ZFPs mediate various biological functions [25]. Apart from their capacity for epigenetic repression of gene promoter regions, the two well-characterized roles are the participation in gene imprinting and silencing of transposable elements (TEs). Genomic imprinting is an epigenetic mechanism responsible for monoallelic gene expression dependently on the parental allele origin [40,41]. The imprinted genes are usually located in clusters, and their exact expression pattern is controlled by the epigenetic marking (e.g., DNA methylation) of imprinting control regions (ICRs). This specific methylation pattern must be preserved during several epigenetic reprogramming events occurring during development. Two KRAB-ZFP factors, namely ZFP57 and ZNF445, are involved in ICR protection. ZFP57, for example, preserves the methylation of most ICRs by binding to methylated hexanucleotide sequence TGCCGC within ICR and subsequently recruiting KAP1, which mediates the DNA methyltransferase recruitment (Figure 3) [42,43,44].

The second well-described phenomenon related to KRAB-ZFP repressive function is the silencing of TEs, which constitute a vast proportion of the human genome [26,45,46]. Retrotransposons are often silenced during early embryogenesis by H3K9 and DNA methylation via KRAB-ZFP/KAP1 repressive complex. Indeed, KAP1 silencing was demonstrated to activate many TE classes. This effect is accompanied by H3K9 demethylation, H4, and H3K27 acetylation, as well as H3K4 and H3K36 trimethylation within affected loci [47,48]. According to Imbeault and colleagues, most of the KRAB-ZFP-binding sequences are located within the TE sequence, while about a third targets other genomic elements, such as promoters or simple repeats [26,27]. In fact, the available data strongly support the hypothesis that TEs and KRAB-ZFPs co-evolved with each other in an “arms race” mechanism. This mechanism may be explained by repeated cycles of TE sequence alterations escaping KRAB-ZFP-mediated repression followed by KRAB-ZFP mutations aimed at the adequate recognition and silencing of the novel TE [49]. TEs, due to their homology, may be vulnerable to recombination, whereas their mobility facilitates integration into new genomic regions, which altogether play a role in genetic diversity, as well as potentially harmful mutations.

Moreover, TEs may serve as a rich source of various cis-regulatory sequences, including enhancers, promoters, suppressors, insulators, and TF-binding sites, thus providing transcriptional and epigenetic control over neighboring genes [50,51]. For instance, the de-repression of certain TE-derived enhancers is crucial for the activation of the embryonic genome in the preimplantation embryos. However, to facilitate an exact transcriptomic control required for further differentiation, these TEs become restricted by their cognate KRAB-ZFPs [52]. Therefore, tight regulation of TE epigenetic status is essential for appropriate development and cell homeostasis, whereas any dysregulation may lead to aberrant differentiation and various pathologic processes, including carcinogenesis [26,45,53,54]. Of note, KAP1-mediated retroelement repression may be hampered due to its inefficient ribosylation. In healthy cells, SIRT6 colocalizes with KAP1 at the 5′UTR of retrotransposon L1 and promotes KAP1 ribosylation. This augments KAP1 interaction with HP1 and favors heterochromatization. Interestingly, upon aging and in response to DNA damage (the processes linked to cancer development), SIRT6 detaches from L1 sequences, which disrupts the KAP1-corepressor complex, and activates L1 expression [55].

## 5. KRAB-ZFPs Roles in Cancer Cells

As mentioned above, the KRAB-ZFP gene superfamily is highly complex, so its involvement in cancer formation remains poorly characterized. Nevertheless, emerging evidence indicates a significant role of KRAB-ZFP proteins in cancer cells. Their expression is perturbed in multiple tumors, as described in more detail below. Many family members are highly overexpressed, whereas the expression of others is reduced (Figure 4). The available literature demonstrates that some of the KRAB-ZFPs play an essential role in many different tumor types. At the same time, the involvement of other factors in cancer biology was reported only in a single tumor. From the pan-cancer perspective, the majority of KRAB-ZFPs with altered expression in cancer are subject to downregulation, while only a small subset is commonly overexpressed in the majority of analyzed tumors compared to normal tissues [23,24]. As may be expected from these observations, certain KRAB-ZFPs may act as TSGs, yet others are characterized as oncogenes (Figure 4).

Overall, KRAB-ZFP-oriented cancer research is focused on their association with clinicopathological features, influence on tumor biology in vitro and in vivo, and molecular mechanisms that may be employed to fulfill their cancer-related function. Of note, the molecular events associated with the functioning of KRAB-ZFPs in cancer reveal both repressive and activating transcriptional regulation at the promoters of various genes, which seems to be dependent on other cofactors interacting with a given KRAB-ZFP. Occasionally, KRAB-ZPFs appear as well as scaffolds enhancing the interaction between multiple proteins, which may result in their enzymatic modification, and, as such, in altered functioning. The documented KRAB-ZFP features related to clinical implications, phenotypic studies, and molecular targeting of cancer-related signaling pathways will be summarized in detail in the subsequent sections.

### 5.1. KRAB-ZFPs with Tumor Suppressor Function

#### 5.1.1. ZNF471

The *ZNF471* promoter region occurs frequently methylated in various cancer tissues, including colorectal [56], tongue squamous cell [57], breast [58], gastric [59], and esophageal cancer (Figure 4) [60]. ZNF471 transcript may also be downregulated by the oncogenic miR-942-5p, as shown in non-small cell lung cancer (NSCLC) [61]. The available experimental data indicate that ZNF471 acts as a tumor suppressor in these cancers through its involvement in the regulation of cell apoptosis, migration, invasion, and proliferation. For example, in esophageal squamous cell carcinoma (ESCC), ZNF471 reduces cell growth via G0/G1 arrest (Figure 5), induces apoptosis, and reverses epithelial-to-mesenchymal transition (EMT), resulting in the inhibition of cell invasion and migration. The suppressive role of ZNF471 in ESCC is mediated through the regulation of various tumor suppressors, such as *MAPK10*/*JNK3* and *protocadherin* (*PCDH*) genes that are involved in apoptosis induction and cell adhesion, respectively. Specifically, ZNF471 binds directly to *MAPK10* promoter (Figure 6A) to enhance its expression. MAPK10 in turn activates the proapoptotic program by the induction of transcription and protein cleavage of caspases and PARP [60].

Similarly, upregulated expression in gastric cell lines suppressed cell proliferation, migration, and invasion and resulted in enhanced apoptosis. Cao and colleagues demonstrated that the underlying mechanism of ZNF471 TSG potential involves the promoter binding and H3K9me3-mediated transcriptional repression of *plastin 3* (*PLS3*) and *transcription factor AP-2 alpha* (*TFAP2A*) oncogenes [59]. In breast cancer (BRCA) cell lines and tissues [58], the experimental overexpression of ZNF471 also reduced cell growth, both in vitro and in vivo, induced cell cycle arrest at the S phase (Figure 5) enhanced apoptosis, and attenuated metastasis. Such an effect was associated with the inhibitory influence of ZNF471 on EMT, cell stemness, as well as Wnt/β-catenin and AKT signaling pathways [58]. The Wnt/β-catenin pathway is responsible for cell proliferation, lineage commitment during embryonic development [62], regulation of migration, apoptosis, stem cell renewal, and genetic stability [63]. Therefore, Wnt/β-catenin signaling is strongly linked to tumor biology, and in particular, to a more aggressive phenotype [63]. During the canonical Wnt/β-catenin pathway activation, Wnt binds to the Frizzled receptor and its co-receptor low-density lipoprotein receptor-related protein 6 (LRP6), which together mediate the stabilization and nuclear translocation of the transcriptional activator, β-catenin. The activated β-catenin induces its downstream target genes (e.g., pro-proliferative c-MYC and cyclin D1 (CCND1), or migration-related matrix metalloproteinases: MMPs) [62,64]. Interestingly, many cancer-related KRAB-ZFPs with TSG function appear as regulators of the Wnt/β-catenin pathway.

#### 5.1.2. ZNF382

ZNF382 also functions as a tumor suppressor, and its expression may be silenced due to promoter hypermethylation. ZNF382 was shown downregulated in multiple cancer types, including leukemia and the solid tumors of head and neck, lung, esophagus, colon, stomach, cervix, and breast (Figure 4) [65,66,67,68]. The suppressive effect of ZNF382 is mediated via inhibition of the NF-κB and AP-1 signaling [65]. NFκB family is involved in many processes in the human organism, such as inflammation, immune response, cell proliferation, promotion of cell cycle, apoptosis inhibition, and cell adhesion [69,70]. Indeed, ZNF382-related disturbance of the pathway evokes cell apoptosis and impedes proliferation or survival. Moreover, the ZNF382 factor is responsible for the downregulated expression of many oncogenes (e.g., *MYC*, *MITF*, *HMGA2*) and NF-κB upstream factors (e.g., *STAT3*, *STAT5B*, *ID1*). The suppressing mechanism of ZNF382 was proposed to be achieved through heterochromatin formation [65]. In ESCC, ZNF382 reduces cell proliferation through increased G2/M cell cycle arrest (Figure 5) and apoptosis and hinders migratory, invasive, and metastatic potential (Figure 6A) [68]. Mechanistically, ZNF382 binds and represses the promoters of *FZD1* (Frizzled Class Receptor 1) and *DVL2* (Dishevelled Segment Polarity Protein 2), essential factors for β-catenin activation. This results in decreased β-catenin activity, and subsequently, reduced expression of its target genes, including *MYC*, *CCND1*, *MMP2*, and *MMP3* (Figure 6A) [68].

#### 5.1.3. ZNF545

*ZNF545* is another tumor-related gene, whose expression may be silenced by methylation, as shown in multiple myeloma [71], as well as in gastric [72], esophageal [73], colorectal [74], breast [75], and liver [76] cancers (Figure 4). In general, the in vitro phenotypic assays in various cancer models indicate that the restoration of ZNF545 expression may result in higher apoptosis and cell cycle arrest (Figure 5), as well as reduced proliferation, migration, and invasiveness. In addition, in vivo data demonstrate decreased tumor growth upon reexpression of *ZNF545*. Together with the observation that *ZNF545* promoter methylation correlates with poorer clinical outcomes, these data highlight the TSG function of ZNF545 in multiple malignancies [72,74,75,77].

Nevertheless, the molecular mechanisms behind such an influence on tumor cell behavior may vary depending on the cancer type. For example, in the breast cancer cell line, a high level of ZNF545 was associated with attenuated proliferation, cell cycle G0/G1 arrest (Figure 5), apoptosis induction, and increased expression of c-Jun/AP1, BAX, p53, and Caspase 3 [75]. The causative role of ZNF545 in p53 transcriptional activation was also observed in multiple myeloma [71]. Additionally, ZNF545 was shown to downregulate crucial cancer-related signaling pathways, such as Wnt/β-catenin, PI3K/AKT, MAPK/ERK [74], NF-κB [77,78], and AP1 [78]. In gastric and colorectal cancer cell lines, ZNF545 was confirmed to be located in the nucleus with the apparent accumulation in the nucleoli. In these cell lines, ZNF545 mediated its function via binding to the rDNA promoter and inhibiting rRNA transcription [72,78]. Wang and colleagues demonstrated that ZNF545 induces changes within histone PTM profile within rDNA promoter, leading to a reduced level of activating H3K4me3 mark. Moreover, ZNF545 was shown to recruit its co-repressor—heterochromatin protein HP1β. HP1β is responsible for binding to methylated H3K9, thus contributing to heterochromatin’s maintenance [72]. Nevertheless, more data are needed to fully elucidate the molecular involvement of ZNF545 in carcinogenesis.

#### 5.1.4. ZNF331

*ZNF331*, located on chromosome 19q13.4, is also known as *Rearranged in Thyroid Adenomas* (*RITA*), *ZNF361*, and *ZNF463* (HGNC: 15489). Of note, *ZNF331* should not be mistaken with another gene known as *RITA*, namely *RBPJ interacting and tubulin associated 1* (HGNC: 25925). The *ZNF331* promoter is commonly methylated in several malignancies, including colorectal [79,80], esophageal [81], gastric [82,83], pancreatic, and cholangiocarcinoma (Figure 4) [82]. Additionally, a somatic mutation within *ZNF331* was identified in follicular thyroid cancers; however, the functional characterization of this variant was not performed [84]. In colorectal tumors, *ZNF331* methylation was associated with short overall survival (OS) [79,80], disease-free survival (DFS), and larger tumor size [79]. Moreover, *ZNF331* promoter methylation correlated with CpG Island Methylator Phenotype (CIMP) and its markers: *MLH1* promoter methylation and *BRAF* mutation [80]. However, the observed linkage between *ZNF331* methylation and CIMP features was contradictory to the similar analysis conducted by Wang and colleagues, in which no such association was noted [79].

Immunohistochemical analysis in gastric cancer tissues revealed that ZNF331 has mainly cytoplasmic localization in tumors, while in normal cells, it resides within the nucleus [83]. According to the studies mentioned above, low ZNF331 expression in gastric cancer correlated with advanced TNM stage, diffuse cancer type, and poorer survival [83]. In the in vitro experiments, ectopic ZNF331 expression resulted in inhibited cell proliferation, cell cycle arrest at the G1/S phase (Figure 5), reduced colony formation, migration, and invasiveness, while apoptosis status remained unchanged [79,83]. In addition, the inhibitory effect on cell migration and invasion, but not on proliferation and cell cycle, was confirmed in the esophageal cancer model [81]. The exact molecular mechanisms promoting its TSG role are still poorly understood. However, ten downstream targets of ZNF331 were identified in gastric cancer based on the comparative 2D gel assay followed by mass spectrometry. These included the genes promoting cell growth, metastasis, gene stability, and cell senescence. The function of one of them (actin-depolymerizing factor: Destrin, DSTN) in promoting cell colony formation and migration was successfully validated [83].

#### 5.1.5. ZNF23

*ZNF23* is located within the 16q22 region, which is frequently changed in several tumors. ZNF23 contains a truncated KRAB-A domain; however, the data suggest that the ZNF23 inhibitory effect does not rely on the KRAB domain but on its last four zinc fingers on the C-terminal end [85]. ZNF23 was found to be downregulated in hepatocellular carcinoma [86], ovarian cancer [85,87], endometrial cancer [85], melanoma (Figure 4) [88], as well as in various tumor cell lines (endometrial, liver, glia, neuroblast, ovary) as compared to normal controls [85]. It is still unclear how ZNF23 is silenced. Huang’s group speculated that it might be due to promoter hypermethylation or loss of heterozygosity that frequently occurs within the ZNF23 location in various malignancies [85]. The clinicopathological analysis in cutaneous melanoma showed that low ZNF23 expression correlates with poor prognostic factors, such as higher tumor thickness, lymph node metastasis, and shorter overall survival [88]. In the in vitro assays, ZNF23 exhibited typical TSG properties, as its overexpression resulted in cell growth suppression [85,87,88], cell cycle arrest in the G1 phase (Figure 5) [85,87], increased apoptosis [87,88], and reduced migration and invasiveness [88].

The ZNF23-induced decrease in proliferation and increase in apoptosis was ascribed (at least partially) to the upregulation of p27 (Figure 6A) [85,88], while p53, CCND1, Rb, and p21 were not affected by ZNF23 overexpression [85]. Apoptosis induction by ZNF23 was also associated with caspase-3 and -9 activation and Cytochrome C release due to the loss of mitochondrial membrane potential [87,88]. Additionally, the antiapoptotic BCL-XL [87] and BCL2 [88] factors involved in inhibiting mitochondrial-mediated apoptosis were demonstrated to be downregulated in ZNF23-overexpressing cell lines. However, the exact mechanism in which ZNF23 reduces BCL genes at the mRNA and protein level remains unknown [87,88]. In melanoma, Zhang and colleagues showed that ZNF23 exerts its TSG functions not only via mitochondrial-related apoptosis and caspase-3 cleavage but also through p53 upregulation [88]. Interestingly, a study in the HepG2 cell line showed that ZNF23 expression positively correlates with cisplatin-induced apoptosis in a dose-dependent manner. However, the cause and effect mechanisms of this phenomenon were not examined in more detail [86].

#### 5.1.6. ZBRK1

ZBRK1 factor (HGNC:16656; zinc-finger and BRCA-1 interacting protein with KRAB domain) is also known as ZNF350. It contains a KRAB domain at N-terminal and eight C2H2 zinc fingers at C-terminal, including a homo-oligomerization subdomain (zinc fingers 5–8), known as BRCA1-dependent transcriptional repression domain (CTRD). The CTRD domain binds to a canonical motif GGGxxxCAGxxxTTT on target genes and functions as a transcriptional repressor [89,90]. Lin and colleagues indicated that ZBRK1 is significantly downregulated in cervical cancer cells, whereas its overexpression in the HeLa cell line results in suppressed tumor growth and inhibits cell migration [91]. Moreover, the promoter of metastatic gene *MMP9* was identified as a direct target for ZBRK1 (Figure 6A), and its expression is inversely correlated with ZBRK1 in cervical cancer clinical samples. Together, these results suggest the importance of ZBRK1 as a tumor suppressor, particularly in the case of metastatic processes through regulating MMP9 expression (Figure 6A) [91]. The same group also identified a correlation between high KAP1 expression and increased migration and invasion of HeLa cells, demonstrating the ability of KAP1 to promote metastasis. ZBRK1 may suppress KAP1 expression by direct binding to its promoter. Interestingly, the repression is dependent on the interaction between ZBRK1 and BRCA1. Thus, the loss of ZBRK1 observed in cancers leads to enhanced metastatic potential induced by KAP1 overexpression [92].

Recently, two additional ZBRK1 target genes, which are implicated in cellular metabolism, were found to be directly bound and repressed by ZBRK1/BRCA1 transcriptional repressor complex: *Phosphofructokinase P* (*PFKP*) [93] and *Glutamate-oxaloacetate Transaminase 2* (*GOT2*) [94]. Both genes were shown overexpressed in BRCA, causing glycolysis dysregulation and increased aspartate metabolism, respectively. Reduced ZBRK1/BRCA1 activity on their promoters resulted in PFKP and GOT2 upregulation, which was associated with adverse outcomes in BRCA patients [93,94]. BRCA1/ZBRK1 duet was shown as well to interact with another transcriptional co-repressor, namely CtBP-interacting protein (CtIP). The resulting complex localizes within various gene promoters and inhibits their expression. For example, angiopoietin-1 (ANG1), an angiogenesis promoting factor, was demonstrated to be activated after removing any partners from the repressive complex. This activation led to enlarged tumor blood vessels and accelerated tumor growth, as observed in a BRCA1-deficient mouse model [95]. Similar observations were described for *HMGA2*, an oncogene involved in the induction of carcinogenesis and metastasis progression. BRCA1/ZBRK1/CtIP complex repressed *HMGA2* promoter in mammary epithelial cells, which led to enhanced proliferation and mammary tumorigenesis [96]. Apart from BRCA1, ZBRK1 may also form a complex with Von Hippel–Lindau (VHL) tumor suppressor, and dependently on other interacting proteins, it may either repress or induce the expression from the bound promoters [97]. For instance, ZBRK1/VHL complex was shown to inhibit VEGF expression, whereas ZBRK1/VHL/p300 induced VHL expression. In the same report, the phenotypic assays confirmed that the enhanced expression of ZBRK1 inhibits renal cancer growth in xenograft models and reduces the cell growth, migration, and invasion capability of renal cancer cell lines [97].

#### 5.1.7. APAK

APAK (ATM and p53-associated KZNF protein; HGNC:20649), also known as ZNF420, consists of the KRAB box domain and 19 consecutive C2H2 zinc fingers. Several studies demonstrated that this protein participates in the inhibition of p53-mediated apoptosis, and its activity is regulated by different mechanisms dependently on the type of stress factor [98,99,100,101]. In unstressed cells, APAK directly binds p53 protein to mediate its deacetylation by recruiting KAP1 and histone deacetylase 1 (HDAC1) complex. Loss of acetylation destabilizes p53, thus hampering its transcriptional activity and promoting degradation. Altogether, p53 downregulation leads to the selective reduction of proapoptotic genes, and ultimately, to the negative regulation of apoptosis. In addition to the deacetylation mechanism, APAK interacts with ATM kinase, a p53 activator, to suppress p53 functioning. In response to DNA damage, ATM phosphorylates APAK on Ser68 (Figure 6A). Phosphorylated APAK separates from p53, which subsequently results in p53 activation and apoptosis induction (Figure 6A) [98]. Another experimental data indicated that APAK competes with p53 in binding to the intron 1 of the proapoptotic *p53AIP1* gene, suggesting an additional alternative mechanism of APAK’s ability to inhibit p53-mediated apoptosis. DNA damage induces APAK to release from the *p53AIP1*, which allows for p53-mediated transcriptional activation and the onset of apoptosis [100].

The same group also demonstrated that upon oncogenic stress, APAK is negatively regulated by ARF protein instead of ATM. ARF was shown to compete with p53 for APAK-binding sites. In the case of oncogene activation (e.g., due to c-MYC overexpression), the ARF level increases and disrupts the interaction between APAK and p53, allowing for the p53-mediated activation of apoptosis [101]. In addition to the ATM- and ARF-dependent control, APAK may also be regulated through epigenetic repression [99]. Under hypoxic conditions, the level of repressive chromatin mark (H3K9me3) within the *APAK* locus increases, leading to gene inactivation. APAK underexpression evoked by low oxygen levels abrogates the inhibitory effect on p53, which altogether leads to p53-dependent apoptosis [99]. Hypoxia is observed in the majority of solid tumors and correlates with poor patient outcomes. Given the fact that epigenetic changes may be relatively easily reversible, i.e., by the administration of small molecule inhibitors, these findings may be of use for the development of future therapies for hypoxic tumors.

#### 5.1.8. ZNF307

ZNF307, also referred to as ZKSCAN4, ZNF427, or ZSCAN36 consists of a KRAB A-box, SCAN domain, and consecutive zinc finger motifs [102]. Li and colleagues demonstrated that ZNF307 upregulates the expression of a negative p53 regulator, MDM2, and histone acetyltransferase EP300, which likely mediates p53 degradation, and, consequently, decreased expression of p53 target genes, e.g., p21. ZNF307 was also shown to repress other cancer-associated pathways (e.g., AP-1, NF-κB), but to a lesser extent than p53 signaling [103]. Another line of research indicated that ZNF307 was significantly downregulated in hepatocellular carcinoma (HCC) tissues (Figure 4). Its restored expression resulted in suppressed cell proliferation, migration, and invasiveness in vitro and reduced tumor formation in a mouse xenograft assay. Moreover, ZNF307 overexpression enhanced apoptosis rate, which was accompanied by higher levels of caspase-3 and BAX, and reduced BCL2 [102].

#### 5.1.9. ZNF496

ZNF496 factor has three different domains: SCAN domain, KRAB domain, and five zinc finger motifs on the C-terminal end. ZNF496 was found downregulated in BRCA tissues (Figure 4), and its low expression correlated with poor survival in BRCA patients [104]. In their in vitro studies, Wang and colleagues demonstrated that ZNF496 inhibits the transcriptional activity of estrogen receptors alpha (ERα) by competitive interaction with its DNA-binding domain. Such a mechanism selectively hinders ERα binding to estrogen-responsive elements in an E2-dependent manner. As a result, ZNF496 suppresses the expression of ERα target genes and reduces breast cancer cell growth (Figure 6A) [104].

### 5.2. KRAB-ZFPs with Oncogenic Properties

#### 5.2.1. RBAK

RBAK (RB-associated KRAB zinc finger) was reported to be upregulated in NSCLC [105], cholangiocarcinoma [106], and prostate cancer (Figure 4), likely due to a decreased level of its upstream negative regulator—miR135a [107]. The initial studies on RBAK provided evidence of its physical interaction with retinoblastoma protein (RB). RBAK seemed to function as an RB co-repressor of proliferation-promoting genes regulated by E2F factors [108]. Apart from its interaction with RB, RBAK may also form a dual complex with RB and androgen receptor (AR). Altogether these interactions lead to augmented AR activity, possibly contributing to the regulation of cell cycle [109]. From the clinical perspective, high RBAK expression correlated with shorter disease-free survival in lung adenocarcinomas (Figure 6B) [105]. The bioinformatics analyses implied that RBAK might partner with a number of cancer-related proteins, including AR, FOXA1, TP53, and E2F transcription factors, which hints towards RBAK involvement in the regulation of transcription. Further functional annotation of genes co-expressed with RBAK identified several biological processes essential for carcinogenesis, e.g., Wnt, NF-κB, and Hippo signaling pathways, mRNA splicing and stability, protein polyubiquitination, DNA repair, chromatin remodeling, cell cycle regulation, and cell adhesion. Together with the in vitro data demonstrating that RBAK-silenced lung cancer cells have lower migration and invasion capacity (Figure 6B), these results implicate RBAK association with the progression of metastases in NSCLC [105]. Furthermore, RBAK was shown to inhibit apoptosis in prostate cancer (Figure 6B). RNAi-mediated silencing of RBAK resulted in impeded cell growth (Figure 6B), an increase of G1 phase, and a decrease in the S phase population, which confirms RBAK participation in the regulation of the cell cycle [107].

#### 5.2.2. P52-ZER6

ZER6, known as well as ZNF398, has two main isoforms: P52-ZER6 and P71-ZER6 [110]. The p52-ZER6 isoform has high expression in colorectal cancer (Figure 4), which coincides with disease progression [111]. Huang and colleagues provided evidence that p52-ZER6 (but not p71-ZER6) enhances the interaction between p53 and MDM2, which in turn augments p53 ubiquitination and degradation. A truncated KRAB domain at the N-terminus of the p52 isoform plays a role in an increased binding between p53 and MDM2. The additional HUB-1 domain in the p71 isoform prevents the factor from enhancing p53/MDM2 partnering. ZER6 knockdown resulted in the increased expression of the p53 target, namely p21 (Figure 6B), which negatively regulates the cell cycle (Figure 5). This further hindered cell proliferation in human colon carcinoma cell line HCT116 [111]. High p52-ZER6 expression was also detected in ER-positive breast cancer cell lines. Moreover, a positive correlation was observed between ZER6 expression and ER expression in primary BRCA [112]. Nevertheless, the exact influence of ZER6 on the development of BRCA remains unknown.

Interestingly, ZER6 appears crucial for cell stemness. It was identified as an essential factor participating in the acquisition and maintenance of self-renewal and epithelial features of pluripotent stem cells (PSC) [113]. In PSCs, TGFβ activates ZER6, which in turn positively influences the expression of pluripotency and epithelial markers, concomitantly inhibiting the expression of differentiation and mesenchymal genes. Specifically, ZER6 colocalizes with SMAD3 and histone acetyltransferase EP300 at active enhancers and promoters to mediate the transcription of TGFβ targets [113]. It remains to be tested whether ZER6 may also be of similar importance in cancer stem cells.

#### 5.2.3. ZNF300

The first studies on ZNF300 indicated its repressive activity and nuclear localization [114]. The available data point toward ZNF300 contribution to immune cell development, as well as to carcinogenesis and cancer-related inflammatory signaling [115,116,117]. ZNF300 binding to DNA overlaps with early growth response 1 (EGR1)-binding sites, and both factors were demonstrated to co-stimulate the activity of Interleukin-2 Receptor-β (IL-2Rβ) promoter [117]. Moreover, Cai and colleagues showed that ZNF300 overexpression occurs and is essential in the leukemic K652 cells undergoing megakaryocytic and erythrocytic differentiation. The study also revealed that ZNF300 silencing promotes cell proliferation, an increase in S phase population (Figure 5), overexpression of PCNA proliferation marker, decreased expression of the cell cycle controllers: p15 and p27, as well as repression of MAPK/ERK pathway [115]. In the HeLa cell line, ZNF300 expression was also shown to be induced by inflammatory factors, which resulted in NF-κB pathway activation. Specifically, ZNF300 stimulated MAPK/ERK signaling, decreased p21 and p27, and mediated the secretion of IL-6 and IL-8, known for promoting inflammation and tumor progression. The functional assays indicated that ZNF300 promotes cell proliferation, colony formation in vitro, as well as tumor growth in a xenograft mouse model [116].

Slightly contrasting data have been reported in NSCLC by Yu and colleagues [118]. ZNF300 overexpression was identified in lung cancer cell lines resistant to chemotherapy, whereas its high levels protected from cisplatin treatment in vitro and in vivo. In the clinical specimens, a high level of ZNF300 expression was linked to more advanced and aggressive tumor phenotypes, as well as shorter survival. Of note, the bioinformatics analysis indicated elevated ZNF300 expression in other highly aggressive tumor types, including anaplastic oligodendroglioma, glioblastoma, skin basal cell carcinoma, and invasive breast carcinomas (Figure 4). In the phenotypic studies, the cells with higher ZNF300 levels showed reduced proliferation with the concomitant G2 cell cycle arrest (Figure 5). Furthermore, with the overexpression of ZNF300, the MAPK/ERK-controlled genes became downregulated, while the genes implicated in the cell cycle (p15, p27) and cancer stemness (NANOG, OCT4) became upregulated [118].

#### 5.2.4. ZKSCAN3

Apart from KRAB domains, ZKSCAN3 also has a SCAN domain [119]. ZKSCAN3 expression is elevated and frequently associated with poor prognosis in the tumors of various origin, including colon [120], prostate [121], bladder [122], breast [123], cervix [124], stomach [119], and liver (Figure 4) [125]. As such, ZKSCAN3 may appear as a useful prognostic marker in various malignancies [119,123,126]. ZKSCAN3 exhibits typical features of an oncogene in many cancer models. In colorectal cancer, ZKSCAN3 overexpression increased anchorage-independent growth *in vitro*, orthotopic tumor growth, and liver metastasis in vivo, as well as resistance to the treatment with 5-fluorouracil [120]. In another study on colorectal cancer, ZKSCAN3 expression was interconnected with cancer-related proteins, i.e., carcinoembryonic antigen (CEA) (Figure 6B), VEGF, and AKT levels in the liver metastasis. CEA additionally promoted the invasive properties of ZKSCAN3-overexpressing cells [126]. Furthermore, ZKSCAN3 was shown to increase cancer cell viability, migration, invasiveness (Figure 6B), and in vivo tumorigenicity in prostate, bladder, and breast cancer models [121,122,123].

Although the exact molecular mechanisms conferring ZKSCAN3 function in cancer remain unclear, several downstream signaling pathways were associated with its high expression. For example, ZKSCAN3 recurrently mediates the expression of MMP-2 and -9 [122,123,126]. Additionally, in BRCA, the expression of ZKSCAN3 positively correlates with the expression of CCND1, BCL2 and negatively with the expression of BAX. ZKSCAN3 silencing also inhibited the phosphorylation, and thus, activation of the AKT/mTOR pathway (Figure 6B) [123]. In HCC, ZKSCAN3 mediates EMT, which is evidenced by the specific expression pattern of EMT markers (E-cadherin^low^, N-cadherin^high^, vimentin^high^), increased migration, and invasion in vitro, as well as augmented metastasis in vivo. Mechanistically, ZKSCAN3 was demonstrated to bind to the promoter of *Integrin β4* (*ITGB4*) to induce its expression. ITGB4, in turn, promoted AKT phosphorylation via activation of the non-receptor tyrosine kinase FAK. This signaling cascade was required for efficient EMT [125]. ZKSCAN3 was also shown to interact with the *cyclin D2* (*CCND2*) promoter and activate its expression in multiple myeloma, in which ZKSCAN3 is also highly expressed compared to normal samples (Figure 4) [127]. In summary, these observations highlight a promoting role of ZKSCAN3 in tumor progression and metastasis, resulting in poor patient outcomes, which renders it a possible therapeutic target.

#### 5.2.5. ZNF10

ZNF10, known as well as KOX1, contains KRAB-A and KRAB-B domains followed by 11 zinc finger motifs. Overexpression and oncogenic characteristics of this KRAB-ZFP gene were reported in breast invasive ductal cancer (Figure 4) [128]. In the clinical specimens, its high expression was more frequent in younger patients (≤50), lower tumor stage (T1-T2), and was associated with the BRCA molecular subtypes. In addition, ZNF10 overexpression positively correlated with the expression of β-catenin. In the in vitro assays, ZNF10 knockdown inhibited proliferation, colony formation, cell cycle progression (Figure 5), migration, and invasion of breast cancer cells. Moreover, it increased the percentage of apoptotic cells. Silencing of ZNF10 resulted as well in reduced tumor formation in nude mice. Further studies showed that ZNF10 directly binds to the *β-catenin* promoter and induces its expression. As a consequence, ZNF10 participates in the activation of the β-catenin signaling pathway, which was evidenced by increased levels of its downstream target genes, including the expression of c-Myc, CCND1, and MMP9, as well as GSK3β phosphorylation (at Ser9) [128].

#### 5.2.6. TIPUH1

TIPUH1 (transcription-involved protein upregulated in HCC), as the name suggests, is a protein that is overexpressed in HCC (Figure 4) [129]. This factor consists of a KRAB-A domain and 12 zinc finger domains and localizes in the nucleus. Silva and colleagues showed that TIPUH1-overexpressing NIH3T3 cells tend to form colonies in soft agar in contrast to anchorage-dependent control cells, while its siRNA-mediated knockdown resulted in impeded proliferation in HCC cell lines. Immunoprecipitation studies allowed for the identification of TIPUH1 interacting partners, which included KAP1 transcriptional co-repressor, as well as hnRNPU, hnRNPF, and nucleolin proteins participate in the post-transcriptional RNA processing. These data imply that TIPUH1 may contribute to the repressed expression of genes associated with apoptosis, cell cycle arrest, or growth suppression. It also suggests that altered expression of TIPUH1 may deregulate the production, maturation, and engineering of mRNAs and/or rRNAs in tumor cells. Nevertheless, no further studies were performed to test these hypotheses [129].

#### 5.2.7. Other Putative Oncogenic KRAB-ZFP Factors with a Limited Amount of Data

A subset of KRAB-ZFP factors was associated with carcinogenesis, however, only in one tumor type. For example, the overexpression of ZNF233 was demonstrated in HCC (Figure 4) [130]. Elevated expression of this factor was associated with higher tumor grade, stage, and a trend of adverse prognosis in HCC patients. In the functional tests, high levels of ZNF233 accelerated cell growth rate by increasing the G1/S transition (Figure 5). Moreover, ZNF233 silencing resulted in hindered tumor growth in nude mice [130]. Although the data suggest oncogenic properties of the factors, the molecular mechanisms behind these observations remain to be elucidated. Another KRAB-ZFP gene, ZNF133, was shown overexpressed in osteocarcinomas (Figure 4) [131]. High levels of ZNF133 have also been detected in irradiated fibroblasts and in childhood cancer survivors that developed second primary cancer. This overexpression was linked to copy number variation events [132]. ZNF133 was also recognized as a factor associated with Alagille syndrome—a genetic disease related to the disruptions in the Notch signaling pathway that predisposes to pediatric HCC [133]. In addition, a factor with high homology to ZNF133, HKR1 (also known as ZNF875), exhibited increased expression in lung cancer (Figure 4), which was even more pronounced upon the treatment with platinum drugs [134]. High homology to ZNF133 may indicate shared molecular functions of both factors. Although the data imply that HKR1 may have a significant influence on the regulation of lung cancer progression and resistance to platinum drugs [134], no further exploration of this hypothesis was reported.

### 5.3. KRAB-ZFPs Factors with Evidence of Tumor-Promoting and Suppressive Role

#### 5.3.1. ZFP57

ZFP57 is a relatively well-characterized KRAB-ZFP family member that is specific to embryonic stem cells, where it plays a key role in imprinting maintenance during physiological development [135]. While the methylation of imprinted regions is unstable in cancer cells (with the higher frequency of hypermethylation occurrence), imbalanced expression of ZPF57 may have an effect on the epigenetic and transcriptional control of the genes regulated via imprinting. Many of these genes are required for proper growth, metabolism, and differentiation, and as such, may play a vital role in carcinogenesis [4]. The available data indicate both TSG and oncogenic functions of ZFP57 in different cancer models [136,137,138,139]. ZFP57 expression was upregulated in high-grade gliomas compared to low-grade gliomas [137], as well as in breast, esophageal, gastric, and colon cancers (Figure 4) [139].

In the phenotypic assays, ZFP57 demonstrated oncogenic properties, such as the promotion of the anchorage-independent growth of human fibrosarcoma HT1080 cell line and tumor formation in mouse HT1080 xenograft model [139]. The molecular mechanism behind this phenomenon was ascribed to a ZFP57-regulated imprinted gene, *Insulin-like Growth Factor 2* (*IGF2*), which is known for its involvement in the development of various cancers. ZFP57 knockdown impeded the expression of IGF2, which further resulted in a reduced level of phosphorylated AKT oncogene. Moreover, ZFP57 overexpression led to the malignant transformation of immortalized mouse fibroblasts NIH3T3 cell line [139]. Finally, in colorectal cancer patients, high levels of ZFP57 coincided with liver and lymph node metastases, as well as with decreased progression-free survival, which also pinpoints its oncogenic function [138].

In contrast to oncogene-related data, the study by Chen and colleagues documented ZFP57 downregulation in BRCA cell lines and tissues [136]. Their data indicated that ZFP57 overexpression inhibits the proliferation of BRCA cells, both in vitro and in vivo, and this effect was linked to the blockage of the Wnt/β-catenin signaling pathway (Figure 6A). Further experiments revealed that high levels of ZFP57 negatively regulated another imprinted target of ZFP57, namely mesoderm specific transcript (MEST), which in turn led to the inactivation of Wnt/β-catenin signaling. Indeed, the β-catenin activation was restored by ectopic MEST overexpression despite concomitant high ZFP57. This was followed as well by the upregulation of β-catenin targets: MYC and CCND1. Nevertheless, the exact mechanism by which MEST may affect the Wnt/β-catenin pathway remains unknown [136]. Taking into account the observations that ZFP57 regulates multiple imprinted and non-imprinted loci [140,141], it may be hypothesized that also other genes may have a key role in ZFP57-mediated influence on cancer biology.

#### 5.3.2. ZNF224

In carcinogenesis, ZNF224 was assigned a dual role of a tumor suppressor and oncogene. The gene encoding *ZNF224*, located on chromosome 19 in the q13.2 position, may be expressed to two different proteins—ZNF224 and ZNF255, depending on the presence or absence of exons encoding the KRAB domain, respectively. ZNF224 factor possesses KRAB-A and KRAB-B domains, which confer its ability to repress gene transcription [142]. Its function as an oncogene was discovered in bladder cancer (Figure 4) based on its interaction with DEPDC1 oncogene [143]. The ZNF224/DEPDC1 complex acted synergistically to enhance cell proliferation, whereas the chemical disruption of this interaction (with 11R-DEP:611-628 peptide) induced apoptosis and suppressed the growth of the bladder cancer cells in vivo and in vitro. Further molecular profiling elucidated that the DEPDC1/ZNF224 complex inhibits the transcription of the *A20* gene responsible for the negative regulation of the NF-κB signaling pathway. In this scenario, DEPDC1/ZNF224 complex promoted the nuclear transport of NF-kB, resulting in reduced apoptosis [143].

This thesis was further supported by the report from Li and colleagues, who demonstrated a similar mechanism of the DEPDC1/ZNF224 complex in an HCC cell line (HepG2) [144]. Oncogenic features of ZNF224 have also been documented in BRCA [145] and chronic lymphocytic leukemia (CLL) (Figure 4) [146], specifically in terms of increased chemoresistance against camptothecin and fludarabine, respectively. In BRCA, overexpression of ZNF224 increased colony formation capacity and cell growth via oncogenic miR-663a. ZNF224 was shown to recognize two consensus DNA sequences within the miR-663a promoter and activate its expression in a dose-dependent manner. Mir-663a, in turn, downregulates the expression of p53 and p21 (Figure 6B), which altogether mediate cellular resistance to apoptosis [145]. In contrast, chemoresistance in CLL mediated by ZNF224 was linked to its involvement in cell cycle control. In the study published by Constanzo/Cesaro group [146], ZNF224 was shown to recognize the *cyclin D3* (*CCND3*) promoter and induce its transcription. Indeed, the silencing of ZNF224 reduced the number of cells in the G2/M phase and led to the accumulation of cells in the G0/G1 phase (Figure 5), emphasizing that ZNF224 is an essential factor supporting cell cycle progression [146].

Another set of publications from the Constanzo/Cesaro group points towards ZNF224 TSG function, particularly in the context of increased sensitivity to drug treatment in human leukemias [142,147,148,149,150,151]. Their data demonstrated that several signaling axes might be responsible for such a phenomenon. For example, ZNF224 was found to modulate the transcriptional activity of WT1 (Wilms tumor gene 1) at its targets. Specifically, ZNF224 mediates the expression of proapoptotic genes (e.g., VDR, BAX, BAK) and repression of antiapoptotic genes (e.g., BAG3 or A1/BFL1) in the K562 cell line. Additional studies indicated that cytosine arabinoside (ara-C or cytarabine) chemotherapeutic used in leukemias augmented the expression of ZNF224, which was accompanied by enhanced apoptosis [147]. In accordance with these observations, ZNF224 was also upregulated in response to other therapeutics (i.e., imatinib and tyrosine kinase inhibitors) [149]. This overexpression was ascribed to the drug-induced signaling cascade. The cascade steps included inhibition of BCR-ABL expression, which in turn led to decreased PI3K/AKT and WT1, a direct repressor of ZNF224 expression. Notably, higher basal ZNF224 expression at diagnosis was associated with improved response to imatinib treatment, suggesting increased apoptosis sensitivity in the patients overexpressing ZNF224 [149].

In another study, ZNF224 was shown to be recruited by WT1 to *Interferon Regulatory Factor 8* (*IRF8*) promoter to prevent the repressive activity of WT1 over IRF8 and eventually enable the expression of this proapoptotic TSG [148]. ZNF224 has been demonstrated as well as a transcriptional repressor of c-MYC [151] and receptor tyrosine kinase AXL [150], which may confer drug resistance in chronic myeloid leukemia (CML) patients. The inhibition of AXL expression may potentially lead to the inactivation of downstream signaling pathways (e.g., PI3K, MAPK) involved in enhanced proliferation, the survival of CML cells, and improved drug response [150]. Taken together, ZNF224 in CML acts as a tumor suppressor that is able to promote the expression of proapoptotic genes and repress the expression of antiapoptotic genes to enhance drug resistance. Thus, restoration of ZNF224 expression was proposed as a promising tool that may overcome drug resistance in CML [142,147,148,149,150,151]. Nevertheless, as ZNF224 may confer both oncogenic and suppressive function, probably dependently on the cellular context, its exact role and potential utility in the clinics remains to be tested in more detail.

#### 5.3.3. ZNF268

ZNF268, another KRAB-ZFP with oncogene and TSG potential, was found upregulated in cervical [152] and ovarian [153] cancers (Figure 4). ZNF268 gene may be expressed to a number of splicing isoforms with different expression profiles depending on the tissue type [152,154]. For example, ZNF268b2 was the dominant protein in squamous cervical cancer samples [152], while ZNF268a expression was low, which means that the imbalanced relative level of these protein isoforms in tissue may lead to carcinogenesis. ZNF268-silenced cells showed decreased expression of cyclin D1 and E2 (Figure 6B), cell cycle arrest at the G1/S transition (Figure 5), low proliferation rate, and colony formation ability, as well as augmented sensitivity to apoptosis upon TNFα treatment. Furthermore, ZNF268 knockdown impeded tumor growth and increased the expression of apoptosis markers in a nude mouse model. The data reported by Wang and colleagues suggest that ZNF268b2 oncogenic properties are mediated via interaction with known NF-κB inducers, namely IKKα and IKKβ, thus leading to the activation of the NF-ĸB signaling pathway [152].

Opposing effects upon ZNF268 knockdown were observed in ovarian cancer. Its silencing promoted tumor cell growth in vitro and in vivo by mediating the progression of the cell cycle. However, simultaneously, lowered ZNF268 levels negatively affected cell migration. These data may suggest that ZNF268 positively influences tumor progression by promoting migration. The function of ZNF268 may thus vary depending on the cell type [153]. The inhibitory effect on cancer cell proliferation and concomitant stimulation of migratory potential in ovarian cancer is somewhat confounding in the view of potential oncogenic properties; thus, further studies are needed for an improved understanding of ZNF268 functioning in cancer.

### 5.4. KRAB-ZFP Factors Implicated in Response to Chemotherapy

As described in the sections above, certain KRAB-ZFP factors may be implicated in the cellular response to anticancer chemotherapy. Their expression level may be changed upon drug administration, or their expression and promoter methylation status may constitute a biomarker of treatment response. Moreover, some KRAB-ZFPs may contribute to the cellular pathways conferring chemoresistance or–sensitivity. Currently, an increasing amount of antitumor agents are used in chemotherapy, alone or in combination. Although their overall mode of action is apoptosis induction, an individual therapeutic outcome may be difficult to predict. Moreover, despite many chemotherapeutic agents available on the market, one of the most significant obstacles is chemoresistance. It is a complex process that often involves multiple cellular pathways acting simultaneously [155,156]. Therefore, more research is needed to fully understand this process, and the studies on KRAB-ZFP involvement in both chemoresistance and -sensitivity may provide valuable insights into the treatment response mechanisms.

One of the KRAB-ZFPs implicated in drug response is the abovementioned ZNF224, which constitutes an interesting example due to both oncogenic and TSG roles. In CLL, in which ZNF224 confers apoptosis resistance, the fludarabine treatment was shown to suppress ZNF224 expression, thus increasing apoptosis rates in CLL cell lines [146]. The same pattern was observed in breast cancer treated with camptothecin [145]. On the contrary, in CML, where the expression of ZNF224 is usually silenced, the treatment with ara-C, imatinib, and tyrosine kinase inhibitors increased ZNF224 expression, resulting in enhanced apoptosis, better treatment response, and improved patient prognosis [147,149,150]. Taken together, ZNF224 expression seems to be a promising biomarker of therapeutic outcome. However, a better understanding of the ZNF224 mode of action is necessary due to its dual function [145,147,149,150].

Furthermore, high expression of another KRAB-ZFP, ZNF300, was associated with chemoresistance to cisplatin, gemcitabine, paclitaxel, docetaxel, and pemetrexed in NSCLC [118]. Cisplatin administration induced the expression of ZNF300, leading to augmented chemoresistance. The ZNF300-mediated cisplatin-resistance occurred through downregulation of MAPK/ERK pathways and activation of CDK1, which resulted in inhibited cell growth and slower cell cycle, respectively [118]. In addition, ZNF23 [86] and HKR1 [134] were found upregulated upon treatment with platinum-based drugs in the liver, and lung cancer, respectively. However, no further mechanistic exploration of these phenomena has been reported. Cisplatin-treatment-induced ZNF23 in a dose-dependent manner in liver cancer that usually retains lowered ZNF23 expression. As could be expected, this dose-dependent upregulation correlated with enhanced apoptosis [86]. Based on the data obtained from other tumor types, it may be speculated that increased ZNF23 may contribute to chemosensitivity via apoptosis induction related to the downregulation of antiapoptotic BCL factors, depleted mitochondrial membrane potential, caspase cascade activation, and p53 upregulation [87,88]. Cisplatin chemosensitivity was also demonstrated in the case of ZNF587B [157,158]. The basic expression of ZNF587B was shown to be downregulated in ovarian cancer, where it functions as a tumor suppressor [157]. Its positive influence on cytotoxicity upon cisplatin treatment was demonstrated via genome-wide screening with the CRISPR-Cas9 knock-out library [158], which was further validated in the in vitro functional assays [157]. Apart from these examples, other KRAB-ZFPs may also modulate the cellular response to drug treatment (e.g., RBAK, ZKSKAN3, presumably other KRAB-ZFPs), as many of them influence various signaling pathways involved in proliferation, cell cycle, and apoptosis.

## 6. Conclusions and Future Directions

The increasing amount of data demonstrates that KRAB-ZFP factors play an essential role in cancer biology, both as oncogenes or tumor suppressors. As such, several KRAB-ZFPs were proposed as molecular biomarkers or drug targets to improve the diagnosis, prognosis, and treatment of cancer patients. To date, however, no further clinical exploration of these ideas was undertaken. Given that the expression of many other members of this gene superfamily is deregulated in single or multiple cancer types [23,24], it may be envisaged that also novel factors with yet unknown function may join the group of KRAB-ZFPs implicated in carcinogenesis. Nevertheless, additional phenotypic validation is needed to test this hypothesis. The experimental validation of the KRAB-ZFPs function is particularly important due to the similarities between various homologous factors. For instance, as described above, many different KRAB-ZFPs affect the WNT/β-catenin signaling pathway. Furthermore, specific KRAB-ZFP proteins (e.g., ZBRK1, ZNF589/SZF1) recognize the same or neighbor DNA motifs, which implies that they may regulate common targets [27,159]. Although it remains to be tested whether they confer similar activity over these regions, it is likely that some of the KRAB-ZFPs may have a redundant role. Thus, the loss of one factor may be irrelevant if another one with a similar role remains functional. Due to the high number of reports demonstrating the involvement of KRAB-ZFP in cancer biology, we were not able to cover all described KRAB-ZFP and their function. Nevertheless, we believe that the current review summarizes the most important findings and gives a comprehensive perspective of the enormous transcriptional regulatory potential held by this family.

Interestingly, the results summarized in the 5^th^ section show that the molecular mechanism utilized by a single KRAB-ZFP may be tissue-specific, and its cancer-related functions may be executed by different pathways depending on the cellular make-up. Moreover, certain reports indicate that in contrast to the canonical function of KRAB-ZFPs that is strongly linked to gene repression, many members of the KRAB-ZFP family genes behave as co-activators of gene transcription (as seen, e.g., in the case of ZBRK1, ZNF224) [97,146,148,151]. Such a differential effect seems to be associated with the KRAB-ZFP interactome. It is likely that a KRAB-ZFPs serves as a DNA sequence recognition protein and/or as a scaffold that recruits other cofactors engaged in gene activation.

Although the literature clearly shows the involvement of a number of KRAB-ZFPs in carcinogenesis, and for some of them, certain aspects of their molecular mechanisms were explored, the latter issue requires more attention to improve our understanding of KRAB-ZFP role in cancer. Of note, the canonical role of KRAB-ZFP was rarely tested in the cancer setting. Yet, the aberrant genome-wide imprinting, TE activity, and epigenetic repression of gene promoters play a vital role in tumor biology [4]. The epigenetic de-repression of TE sequences may lead to their mobilization and, as a consequence, to insertional mutagenesis that may disrupt the gene structure, for example, with new coding or regulatory sequences [26,160]. It may as well expose DNA sequences prone to homological recombination, which will additionally contribute to higher mutagenicity. Moreover, it may modify an epigenetic status of a number of regulatory regions located within TE sequences, such as promoters, enhancers, suppressors, or insulators. In this scenario, de-repressed TE sequences may provide a novel platform for TF binding that rewires the transcriptional activity of nearby genes [4,26]. Loss of imprinting (LOI), another frequent event occurring during carcinogenesis, may deregulate the expression from the imprinted regions that harbor genes essential for cell growth, differentiation, and metabolism. All these processes, known to be disturbed in cancer cells, have a vital influence on tumorigenic progression [10,11,161]. Finally, the repression of gene promoters via epigenetic mechanisms potentially mediated by KRAB-ZFPs were either not tested or studied in a single-locus fashion. In the latter approach, the putative genome-wide data for KRAB-ZFP-binding sites were not gathered for a given cell type nor analyzed with the usage of publicly available chromatin immunoprecipitation datasets [27,162]. Yet, KRAB-ZFPs may bind to many genomic locations, which frequently correlates with a repressive histone mark, namely H3K9me3 [27]. Therefore, the global distribution of KRAB-ZFP-binding sites, integrated with transcriptomic, epigenomic (DNA methylome, H3K9me3 profiling), and protein interactome, may significantly advance our understanding of KRAB-ZFP function in the context of cancer cells.

## Figures and Tables

**Figure 1 ijms-22-02212-f001:**
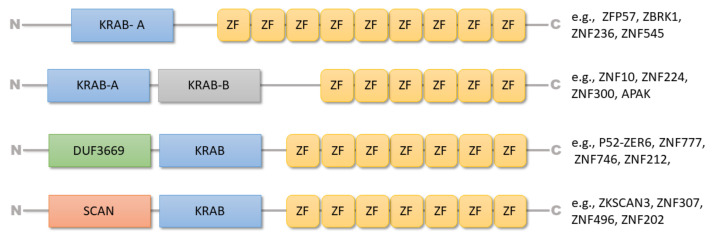
The structure of various Krüppel-associated box zinc finger proteins (KRAB-ZFP) factors and their composing domains. The diagram shows the structure of selected factors (based on the Uniprot database) containing a KRAB domain at the N-terminus and the variable number of zinc fingers at the C-terminus. A KRAB domain may consist of two boxes: KRAB-A and KRAB-B. In some of the KRAB-ZFPs, an additional “SRE-ZBP, CTfin51, AW-1 and Number 18 cDNA” (SCAN) or “domain of the unknown function” (DUF3669) may also be present.

**Figure 2 ijms-22-02212-f002:**
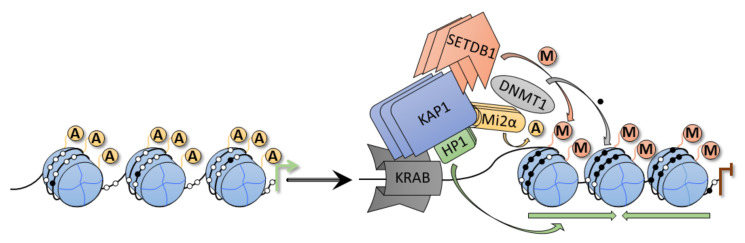
Repressive complex with the participation of a KRAB-ZFP factor. The repression complex contains the scaffold protein KAP1, H3K9 methyltransferase SET Domain Bifurcated 1 (SETDB1), heterochromatin protein 1 (HP1), and repressive Nucleosome Remodeling Deacetylase (NuRD) complex with Mi2α histone deacetylase. In certain cell types, the complex may also interact with DNA methyltransferase. These protein partners cooperatively inhibit transcriptional activity and promoter heterochromatization of the locus recognized by the zinc finger domain within an interacting KRAB-ZFP factor. A—histone acetylation, M—H3K9me3, white circles—unmethylated cytosines, black circles—methylated cytosines.

**Figure 3 ijms-22-02212-f003:**
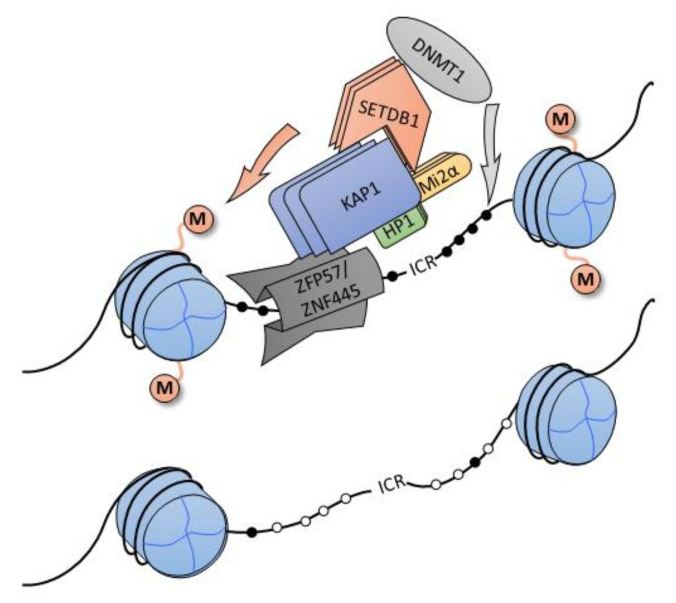
The involvement of ZFP57 and ZFP445 in gene imprinting. Both ZFP57 and ZNF445 are involved in maintaining methylation patterns by recruiting KAP1, DNA methyltransferase DNMT1, and histone methyltransferase SETDB1 to induce H3K9 trimethylation. The methylated imprinting control regions (ICR) on the top (black circles), unmethylated ICR on the bottom (white circles).

**Figure 4 ijms-22-02212-f004:**
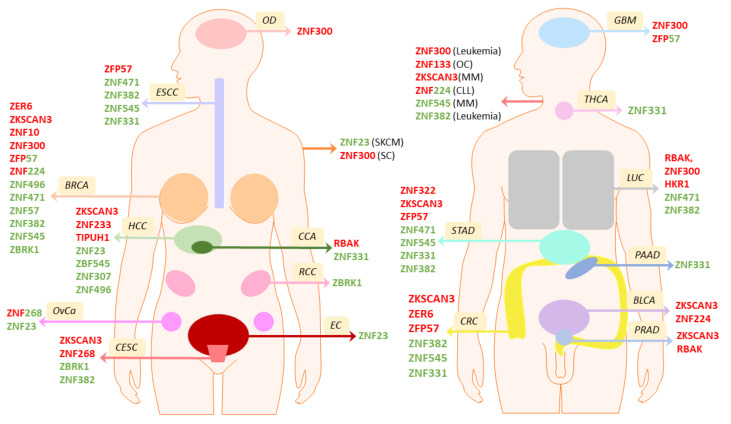
KRAB-ZFPs oncogene and tumor suppressor genes (TSG) role in the cancers of various tissue origins. The scheme presents the localization of different cancers in the human body, in which KRAB-ZFP factors may act as oncogenes (red font, the gene is upregulated) or tumor suppressor genes (green font, the gene is downregulated). Abbreviations: BLCA—bladder cancer, BRCA—breast cancer, CAA—cholangiocarcinoma, CESC—cervical cancer, CLL—chronic lymphocytic leukemia, CRC—colorectal cancer, EC—endometrial cancer, ESCC—esophageal squamous-cell carcinoma, GBM—glioblastoma, HCC—hepatocellular carcinoma, LUC—lung cancer, MM—multiple myeloma, OD—oligodendroglioma; OvCa—ovarian cancer, PAAD—pancreatic cancer, PRAD—prostate adenocarcinoma, RCC—kidney cancer, SC—skin cancer, SKCM—melanoma, STAD—gastric cancer, THCA—thyroid cancer.

**Figure 5 ijms-22-02212-f005:**
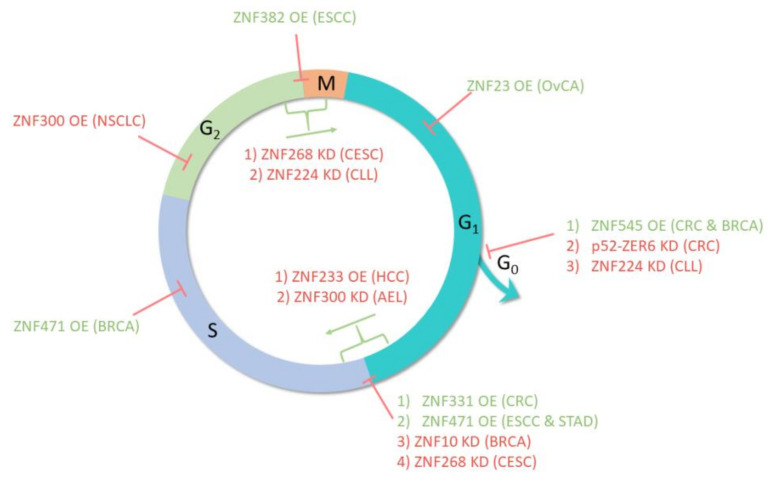
The influence of selected KRAB-ZFPs on cell cycle. The KRAB-ZFP factors may induce cell cycle arrest (red arrows) or promote phase transition (green arrows) in a given phase. Abbreviations: KD—knockdown, OE—overexpression, AEL—acute erythroid leukemia, BRCA—breast cancer, CESC—cervical cancer, CLL—chronic lymphocytic leukemia, CRC—colorectal cancer, ESCC—esophageal squamous-cell carcinoma, HCC—hepatocellular carcinoma, NSCLC—non-small cell lung cancer, OvCa—ovarian cancer, STAD—gastric cancer.

**Figure 6 ijms-22-02212-f006:**
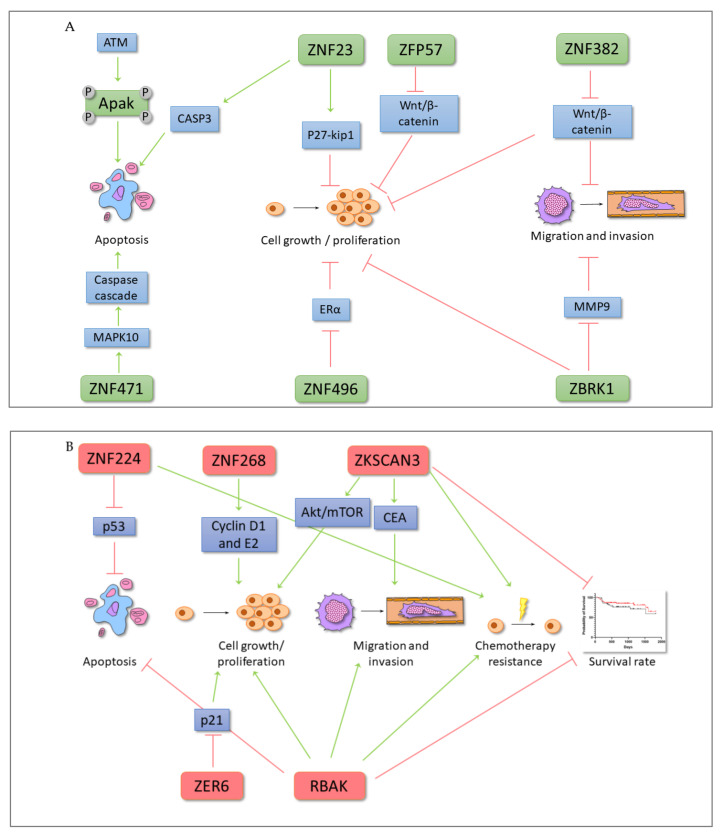
An influence of selected KRAB-ZFP TSGs (**A**) and oncogenes (**B**) on various cancer-related signaling pathways and phenotypic features in neoplastic cells. The schematic representation demonstrates selected KRAB-ZFPs with oncogenic and TSG properties and their impact on the cellular signaling, apoptosis, response to treatment, proliferation, migration, and invasion of cancer cells, as well as patient survival. Some signaling pathways are yet to be understood. Green arrow—induction, red—inhibition, P—phosphorylation.

## Data Availability

Not applicable.

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
