# Peer review of "KRAB-ZFP Transcriptional Regulators Acting as Oncogenes and Tumor Suppressors: An Overview"

_ijms, 2021, doi:10.3390/ijms22042212_

Round 1
Reviewer 1 Report
This is an interesting review focussing on the link between KRAB-ZFP transcription factors and cancer. The manuscript is well written and covers the most recent scientific literature in the field.
Please find my comments below.
figure 1. It is a bit difficult to read the figure after reading the corresponding description in the text. I suggest to modify it by adding a general scheme of the canonical structure of KRAB-FFP proteins followed by the representation of variants. The name of representative known KRAB-FFP proteins associated with a given variant structure may be added next to each variant to improve its readability.
line 163 "...high homology..." Homology is a qualitative (not quantitative) concept. Please remove highly.
lines 166-168. References are missing here, since this is a trend topic in model and non-model organisms. The authors may want to refer to recent reviews by Sundaram et al., 2020 (doi:10.1098/rstb.2019.0347) and Moschetti et al., 2020 (doi: 10.3390/biology9020025) for an up to date description of the contribution of TEs to the host regulatory sequences in mammals and in model organisms.
The paragraph concerning the link between the KAP1 and the activation of transposable elements should be expanded, since this connection is highly relevant . There are many published papers in this field, especially from the Gorbunova group, that should be possibly referenced.
Author Response
We thank the reviewer for helpful comments. We worked on all of them to improve our manuscript. Please find the details about our corrections below.
Reviewer:
This is an interesting review focussing on the link between KRAB-ZFP transcription factors and cancer. The manuscript is well written and covers the most recent scientific literature in the field.
Please find my comments below.
figure 1. It is a bit difficult to read the figure after reading the corresponding description in the text. I suggest to modify it by adding a general scheme of the canonical structure of KRAB-FFP proteins followed by the representation of variants. The name of representative known KRAB-FFP proteins associated with a given variant structure may be added next to each variant to improve its readability.
RE: We corrected Figure 1 accordingly to reviewer's suggestions.
line 163 "...high homology..." Homology is a qualitative (not quantitative) concept. Please remove highly.
RE: We agree with the comment. The "high" word was removed.
lines 166-168. References are missing here, since this is a trend topic in model and non-model organisms. The authors may want to refer to recent reviews by Sundaram et al., 2020 (doi:10.1098/rstb.2019.0347) and Moschetti et al., 2020 (doi: 10.3390/biology9020025) for an up to date description of the contribution of TEs to the host regulatory sequences in mammals and in model organisms.
The paragraph concerning the link between the KAP1 and the activation of transposable elements should be expanded, since this connection is highly relevant . There are many published papers in this field, especially from the Gorbunova group, that should be possibly referenced.
RE: As both comments are related to the Section 4, we made the appropriate corrections based on the reviewer's comment. We included the mentioned citations. Moverover, we expanded the section to include more details on the KAP1 involvement in TE regulation. The changes may be seen in the second and third paragraph of the Section 4: we used the "track changes" option in the text processing software to improve the read-out of all modifications.
Reviewer 2 Report
The paper does a deep literature revision concerning the KRAB-ZFP transcriptional regulators in cancer. They analyze every single reported gene of the Krab-ZFP superfamily and the role in cancer cells.
Even if the article is well structured and face to an actual and important topic I consider fundamental for the pubblication, a paragraphs reorganization, in light of the pharmacology response (where reported) of the decribed genes.
Therefore I suggest adding a paragraph in which the authors describe the effects that chemioterapic or other drugs have on the expression of the treated genes. It would also be interesting that where the literature does not give information about it, speculations could be made on the type of effect that these drugs can trigger in consideration of the molecular pathway in which they are involved. I believe that this information can raise the level of the paper and provide very interesting for readers.
Author Response
We are very grateful for the insightful suggestions from the reviewer. Based on the comment, we included a Section 5.4 that summarizes the available evidence on the involvement of KRAB-ZFP factors in response to chemotherapy.
Round 2
Reviewer 1 Report
The authors have satisfactorily responded to my comments.
Some additional minor corrections should be made before publications.
Some examples are reported below (but please check for additional typos).
line 100. Uniprot should be UniProt
Figure 5 I can't see any red arrow in the figure. Please revise the figure legend's text.
line 635. ", however, only in 635 one tumor type " change to ", even if in a single tumor type"
Reviewer 2 Report
I'm satisfied with the change made by the authors.